# Structure and inhibition of *Cryptococcus neoformans* sterylglucosidase to develop antifungal agents

Nivea Pereira de Sa [1], Adam Taouil[2], Jinwoo Kim[2,3], Timothy Clement[2,3], Reece M. Hoffmann[4], John E. Burke [4,5], Robert C. Rizzo[3,6,7], Iwao Ojima [2,3], Maurizio Del Poeta[1,3,8,9] ✉ & Michael V. Airola [3,10] ✉

Pathogenic fungi exhibit a heavy burden on medical care and new therapies are needed. Here, we develop the fungal specific enzyme sterylglucosidase 1 (Sgl1) as a therapeutic target. Sgl1 converts the immunomodulatory glycolipid ergosterol 3β-D-glucoside to ergosterol and glucose. Previously, we found that genetic deletion of Sgl1 in the pathogenic fungus *Cryptococcus neoformans* (*Cn*) results in ergosterol 3β-D-glucoside accumulation, renders *Cn* non-pathogenic, and immunizes mice against secondary infections by wild-type *Cn*, even in condition of CD4+ T cell deficiency. Here, we disclose two distinct chemical classes that inhibit Sgl1 function in vitro and in *Cn* cells. Pharmacological inhibition of Sgl1 phenocopies a growth defect of the *Cn Δsgl1* mutant and prevents dissemination of wild-type *Cn* to the brain in a mouse model of infection. Crystal structures of Sgl1 alone and with inhibitors explain Sgl1's substrate specificity and enable the rational design of antifungal agents targeting Sgl1.

[1] Department of Microbiology and Immunology, Stony Brook University, Stony Brook, NY, USA. [2] Department of Chemistry, Stony Brook University, Stony Brook, NY, USA. [3] Institute of Chemical Biology and Drug Discovery (ICB&DD), Stony Brook, NY, USA. [4] Department of Biochemistry and Microbiology, University of Victoria, Victoria, BC V8W 2Y2, Canada. [5] Department of Biochemistry and Molecular Biology, University of British Columbia, Vancouver, BC, Canada. [6] Department of Applied Mathematics & Statistics, Stony Brook University, Stony Brook, NY, USA. [7] Laufer Center for Physical & Quantitative Biology, Stony Brook University, Stony Brook, NY, USA. [8] Division of Infectious Diseases, School of Medicine, Stony Brook University, Stony Brook, NY, USA. [9] Veterans Administration Medical Center, Northport, NY, USA. [10] Department of Biochemistry and Cell Biology, Stony Brook University, Stony Brook, NY, USA. ✉email: maurizio.delpoeta@stonybrook.edu; michael.airola@stonybrook.edu

*C*ryptococcus neoformans is a fungal pathogen that, upon entering the lung and disseminating through the bloodstream, causes a life-threatening meningoencephalitis in susceptible immunocompromised patients, leading to high morbidity and mortality[1]. The most widely used antifungal agents target ergosterol synthesis or bind to sterols in the cell wall. For example, azole antifungals (e.g., fluconazole) inhibit cytochrome P450-dependent 14-alpha-sterol demethylase[2], while the polyene antifungal amphotericin B irreversibly binds ergosterol to destabilize the fungal lipid bilayer[3,4]. However, the current armamentarium of antifungals suffer from many drawbacks with amphotericin B and flucytosine being toxic, flucytosine not being available worldwide, echinocandins having a narrow spectrum of activity and not being active against cryptococcosis, and azoles being limited in their use due to drug interaction and resistance[5]. Thus, it is critical to develop new pharmacological agents to combat this life-threatening fungal pathogen.

Sterylglucosides (SGs) are glycolipids produced by plants, fungi, and some bacteria that are derivatives of membrane-bound sterols with a single glucose moiety attached to the 3β-hydroxy group[6]. Typical SGs are ergosterol 3β-D-glucoside (erg-glc) in fungi[7] or sitosterol-3β-D-glucoside in plants[8]. In plants, sitosterol-3β-D-glucoside serves as a primer to initiate the glucan polymerization of cellulose[9]. In fungi, the biological function of erg-glc remains enigmatic, but it accumulates during stress conditions[7] and both erg-glc and plant SGs are known to stimulate immune responses in mammals[6].

*C. neoformans* sterylglucosidase 1 (Sgl1), the first gene/protein identified with a sterylglucosidase activity, converts erg-glc to ergosterol and glucose (Fig. 1a). Sgl1 was originally proposed as a glucosylceramidase due to its ability to utilize fluorescent, short-chain C6-NBD-glucosylceramide (Fig. 1b) as a substrate

in vitro[10]. However, Sgl1 does not hydrolyze long-chain C18-glucosylceramide (Fig. 1c) found endogenously in fungi and genetic deletion of *C. neoformans* Sgl1 causes accumulation of erg-glc, with glucosylceramide levels remaining unchanged[11].

Genetic deletion of *C. neoformans* Sgl1 renders the Δ*sgl1* mutant strain non-pathogenic in a mouse model of cryptococcosis[11]. The Δ*sgl1* mutant fails to disseminate to the brain of infected mice and is cleared from the lung within 2 weeks. Furthermore, mice that recover from *C. neoformans* Δ*sgl1* infection exhibit protective immunity when challenged with the wild-type fungus[11]. Importantly, when mice are CD4+ T cell depleted, which is a hallmark of immunosuppressed HIV patients, the mice still exhibit protective immunity[11]. This suggests that pharmacological inhibition of Sgl1 might similarly be able to induce protective immunity, identifying Sgl1 as a promising therapeutic target for fungal infections.

In this work we determine the structural basis for erg-glc specificity and develop Sgl1 as a therapeutic target for fungal infections. Our analysis of structures of *C. neoformans* Sgl1 reveals a Y-shaped hydrophobic active site pocket that generates specificity for erg-glc and which is not present in human and bacterial glucosylceramidases. We identify two first-in-class and selective small-molecule inhibitors of Sgl1 that cause accumulation of erg-glc in vivo and phenocopy the growth defects of the *Cn* Δ*sgl1* mutant. These unoptimized inhibitors display efficacy in a mouse model of infection and prevent the dissemination of wild-type *Cn* to the brain. Lastly, we present co-crystal structures of Sgl1 with these inhibitors to guide medical chemistry optimization.

## Results

**Substrate specificity.** Full-length *C. neoformans* Sgl1 was produced in *Escherichia coli* and purified using affinity and gel

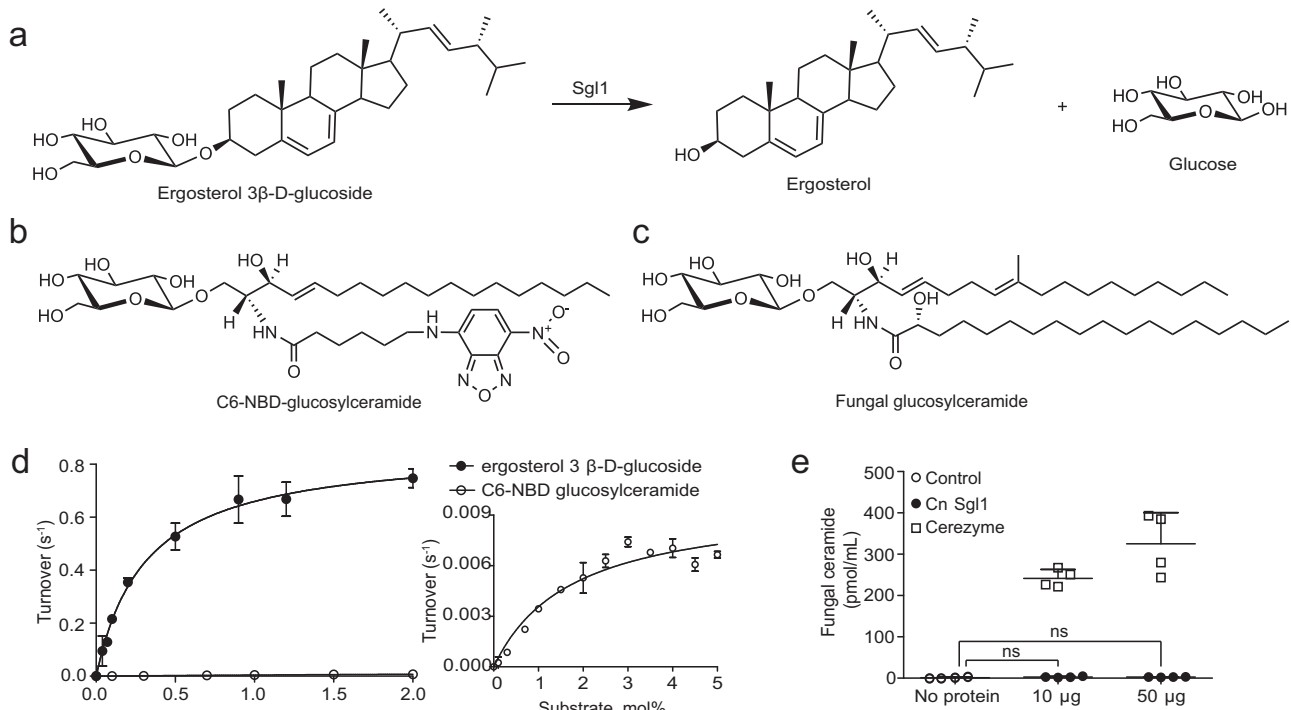

**Fig. 1 Sgl1 is specific for ergosterol 3β-D-glucoside. a** Ergosterol 3β-D-glucoside (erg-glc) hydrolysis by Sgl1 yields ergosterol and glucose. **b** Chemical structures of C6-NBD-glucosylceramide (C6-NBD-GlcCer) and **c** fungal GlcCer. **d** Kinetic analysis of erg-glc (black circles) and C6-NBD-GlcCer hydrolysis (open circles) by Sgl1. Sgl1 readily hydrolyzed erg-glc while C6-NBD-glcCer was a relatively poor substrate. Data represent the mean ± SD for $n = 2$ independent experiments. **e** Comparison of Cerezyme and *C. neoformans (Cn)* Sgl1 activity against fungal long-chain glucosylceramide. There was no significant difference between the control with no enzyme and samples with *Cn* Sgl1 at 10 or 50 µg. Statistical analysis by one-way ANOVA, Dunnett's multiple comparison test. Data represent the mean ± SD of $n = 4$ independent experiments. Source data are provided as a Source Data file.

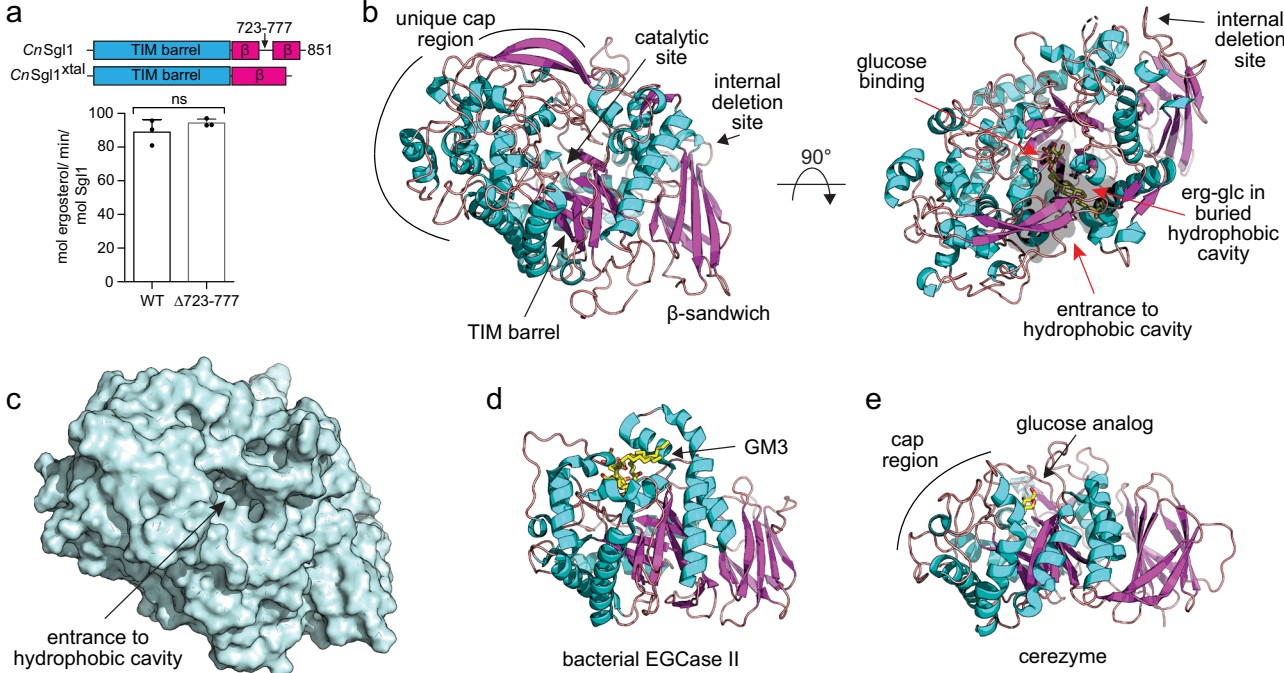

**Fig. 2 Overall structure. a** The deletion of 55 residues in *C. neoformans (Cn)* Sgl1^xtal (Δ723–777) does not affect *Cn* Sgl1 activity towards erg-glc. Data represent the mean ± SD for $n = 3$ independent experiments. Statistical analysis by one-way ANOVA, Dunnett's multiple comparison test. **b** Structure of *Cn* Sgl1. *Cn* Sgl1 is organized in two domains: a TIM barrel containing the catalytic site and a C-terminal β-sandwich domain. A unique cap region encloses the active site to form a Y-shaped cavity (gray outline, visualized using pymol internal surface cavity) for erg-glc binding. Docked erg-glc is shown in yellow with the ergosterol moiety bound in the enclosed hydrophobic cavity. **c** Surface representation of *Cn* Sgl1 showing the entrance to the hydrophobic cavity of the buried active site. **d** Bacterial EGCase II (pdb code: 2OSX) conserves the two domain architecture but has a solvent-exposed active site to hydrolyze gangliosides with more than one sugar. **e** Recombinant human glucosylceramidase, cerezyme (pdb code: 6TJQ) presents similarities with Sgl1, including a small cap region; however, its active site is solvent exposed and does not contain a buried cavity. Source data are provided as a Source Data file.

filtration chromatographies. To establish substrate specificity, we assessed the enzymatic activity of Sgl1 towards erg-glc (Fig. 1a) and C6-NBD-glcCer (Fig. 1b) under similar conditions (Supplementary Fig. 1a–f) and quantified the products by high-performance liquid chromatograph (HPLC). In these experiments, the bulk concentrations of erg-glc and C6-NBD-glcCer were held constant and the molar ratios of erg-glc and C6-NBD-glcCer to Triton X-100 was varied to achieve the indicated surface concentrations expressed as mol%. Sgl1 readily hydrolyzed erg-glc with a $K_M$ of 0.31 ± 0.04 mol% in Triton X-100 mixed micelles and a $k_{cat}$ of 0.62 ± 0.02 s$^{-1}$ (Fig. 1d). In comparison, synthetic C6-NBD-glcCer was a relatively poor substrate with a $K_M$ of 1.73 ± 0.37 mol% and a $k_{cat}$ of 0.0082 ± 0.0003 s$^{-1}$ (Fig. 1d). In line with previous observations[11], we were unable to detect hydrolysis of endogenous fungal glucosylceramide by Sgl1 (Fig. 1c, e). As a positive control, the human glucosylceramidase Cerezyme readily hydrolyzed fungal glucosylceramide (Fig. 1e). Taken together with the previous results that erg-glc, but not fungal glucosylceramide, accumulates in the *C. neoformans Δsgl1* knockout strain, this confirms Sgl1 as the first specific sterylglucosidase.

**Overall structure.** To define the mechanism for ergosterol glucoside specificity, we sought to determine the structure of *C. neoformans* Sgl1. Full-length Sgl1 failed to crystallize. We therefore used hydrogen–deuterium exchange mass spectrometry to identify and then delete internal disordered regions (Supplementary Fig. 2a). Screening of three deletion constructs (Supplementary Fig. 2b) identified one deletion, *C. neoformans* Sgl1 Δ723–777, that did not affect Sgl1 activity towards erg-glc (Fig. 2a) and yielded crystals that diffracted to 2.1 Å resolution

(Table 1). The deleted region between residues 723 and 777 is not conserved among Sgl1 orthologs (Supplementary Fig. 3) and appears specific for *C. neoformans* Sgl1 with no obvious role in Sgl1 function or regulation. Phases were determined using single wavelength anomalous diffraction from selenomethionine-enriched protein (Table 1).

The structure of Sgl1 revealed two domains comprising a catalytic domain with a central TIM barrel and a C-terminal β-sandwich domain (Fig. 2b, c). The general architecture of Sgl1 was most similar to bacterial endoglucoceramidase II (EGCase II)[12] from *Rhodococcus* sp. (Fig. 2d) and the human glucosylceramidase Cerezyme[13] (Fig. 2e), but differed in significant ways. The main differences resided in the larger catalytic domain of Sgl1 that contains 10 β-strands, 28 α-helices, and several ordered loops (Fig. 2b). These additional structural elements combine to form a unique cap region above the TIM barrel that creates an enclosed Y-shaped cavity (Fig. 2b). The active site of Sgl1 is located at the base of the Y-shaped cavity and is capped to limit binding to a single glucose moiety. One arm of the Y-shaped cavity is lined with hydrophobic residues and is buried within the protein structure, which would limit the size of molecules that can fit in this cavity. The other arm is also lined with hydrophobic residues and contains a narrow opening that is solvent-exposed (Fig. 2b, c). These structural features suggest erg-glc can enter the Y-shaped cavity through the solvent exposed opening and bind within the buried hydrophobic cavity. In contrast, the active sites of Cerezyme and EGCase II are completely solvent exposed and do not contain a buried cavity (Fig. 2d, e).

**Active site.** The active site of Sgl1 contained two catalytic glutamate residues found in the canonical topological positions for

**Table 1 Data collection and refinement statistics.**

| | Se-Met | Δ723–777 | Hit 1-Sgl1 complex | Hit 9-Sgl1 complex |
|---|---|---|---|---|
| **Data collection** | | | | |
| Space group | P 1 21 1 | P 1 21 1 | P 1 21 1 | P 1 21 1 |
| Cell dimensions *a, b, c* (Å) | 98.54 133.76 129.91 | 98.60 126.61 126.65 | 98.10 133.16 129.59 | 98.42 133.31 130.30 |
| β (°) | 94.57 | 95.46 | 94.47 | 94.54 |
| Wavelength | 0.979328 | 1.033170 | 0.979321 | 1.033150 |
| Resolution range | 66.87–2.40 (2.44–2.40) | 47.28–2.13 (2.206–2.13) | 58.12–2.85 (2.952–2.85) | 43.3–2.32 (2.403–2.32) |
| Total reflections | 2,733,383 (130,133) | 543,298 (39,020) | 543,767 (55,558) | 482,943 (48,755) |
| Multiplicity | 21.0 (19.3) | 3.2 (2.4) | 7.0 (7.2) | 3.3 (3.4) |
| Completeness (%) | 99.4 (98.8) | 96.54 (85.58) | 96.56 (91.46) | 95.04 (88.72) |
| Mean I/sigma (I) | 12.0 (1.7) | 6.52 (1.57) | 5.10 (1.54) | 6.53 (1.48) |
| Wilson *B*-factor | 39.583 | 23.04 | 35.24 | 25.99 |
| *R*-merge | 0.222 (2.313) | 0.1105 (0.5432) | 0.3334 (1.626) | 0.1519 (0.936) |
| *R*-meas | 0.233 (2.445) | 0.1332 (0.6853) | 0.3601 (1.752) | 0.1812 (1.112) |
| *R*-pim | 0.071 (0.783) | 0.07354 (0.4118) | 0.135 (0.6478) | 0.09787 (0.5951) |
| CC1/2 | 0.997 (0.819) | 0.969 (0.667) | 0.975 (0.701) | 0.989 (0.645) |
| CC* | 0.999 (0.933) | 0.992 (0.894) | 0.994 (0.908) | 0.997 (0.886) |
| **Phasing** | | | | |
| Se sites | 18 | | | |
| Figure of merit | 0.329 | | | |
| **Refinement** | | | | |
| Reflections used in refinement | | 166,716 (14,686) | 74,889 (7080) | 137,536 (12787) |
| *R*-work | | 0.1553 (0.2445) | 0.1934 (0.2687) | 0.1816 (0.2835) |
| *R*-free | | 0.1926 (0.3053) | 0.2309 (0.3388) | 0.2316 (0.3161) |
| CC (work) | | 0.969 (0.872) | 0.944 (0.869) | 0.960 (0.873) |
| CC (free) | | 0.956 (0.792) | 0.916 (0.751) | 0.926 (0.797) |
| Number of non-hydrogen atoms | | 25,317 | 23,938 | 24,988 |
| Macromolecules | | 23,112 | 23,119 | 23,090 |
| Ligands | | 33 | 117 | 123 |
| Solvent | | 2172 | 702 | 1775 |
| Protein residues | | 2909 | 2911 | 2914 |
| RMS (bonds) | | 0.011 | 0.012 | 0.007 |
| RMS (angles) | | 1.03 | 0.97 | 0.69 |

Values in parentheses are for highest-resolution shell.

retaining β-glucosidases (Fig. 3a). The surrounding active site residues for glucose binding more closely resembled bacterial EGCase II (Fig. 3b) than human Cerezyme (Fig. 3c). However, unlike EGCase II, Sgl1 contained a series of capping residues, including a bulky Trp residue that creates a wall to sterically hinder substrates with more than one sugar from binding (Fig. 3a). Cerezyme also contains a capping region that limits hydrolysis of complex glycosphingolipids with multiple sugars, but in Cerezyme, the cap region is much smaller and is formed by a distinct set of residues (Figs. 2e and 3c). The unique active sites of Sgl1 and Cerezyme suggest that selective inhibitors targeting Sgl1 would not inhibit Cerezyme. In support of this, binding of a Tris molecule in the active site of Sgl1 resulted in the Tris molecule adopting a different conformation than that of a Tris molecule previously observed in the active site of Cerezyme[13] (Supplementary Fig. 4).

**Ergosterol 3β-D-glucoside recognition.** Erg-glc has not yet been successfully co-crystallized with either wild type or catalytically inactive Sgl1-E520S. However, computational approaches have been successful in predicting the binding mode of lipid substrates to lipid-modifying enzymes with well-defined hydrophobic pockets[14] like Sgl1. To understand the mechanistic basis for erg-glc substrate recognition and hydrolysis, we used a flexible ligand docking protocol to dock erg-gcl in the Y-shaped pocket of Sgl1. Erg-glc was docked with the glucose moiety in position for nucleophilic attack by the catalytic glutamate residue Glu520 and the ergosterol moiety was surrounded by hydrophobic residues in

the buried hydrophobic arm of the Y-shaped pocket (Fig. 3d). Glucose recognition was mediated by several hydrogen-bond contacts with the residues Lys47, His142, Asn269, Glu270, Trp570, and Glu587 (Fig. 3a, d). Importantly, the glucose moiety was in a similar position to the glucose moiety from co-crystal structures of EGCase II[12] (Fig. 3b) and human cerezyme[13] (Fig. 3c).

Based on the predicted binding pose of erg-glc, we propose that Sgl1 uses a canonical reaction mechanism for erg-glc hydrolysis and a series of polar and non-polar interactions to generate affinity for the glucose and ergosterol moieties of erg-glc. To test this hypothesis, we generated 12 single point mutants of key residues putatively involved in erg-glc recognition and catalysis and tested their activity towards erg-glc and the artificial substrate resorufin-3β-D-glucopyranoside (res-glp). Consistent with our predictions and a prior study of a bacterial endoglucoceramidase[15], Alanine point mutations of residues predicted to directly bind to the glucose moiety (Lys47, His142, Asp144, Tyr453, Trp570, and Glu587), as well as point mutations of the catalytic glutamic acids Glu270 and Glu520, either greatly reduced activity or completely inactivated Sgl1 (Fig. 3e). Except for Glu270, which is the catalytic glutamate that initiates catalysis, all point mutants displayed some residual activity towards either erg-glc or res-glp. This suggests the observed reduction of activity is not due to protein misfolding, but rather to the importance of these residues in substrate recognition and/or catalysis.

Within the hydrophobic cavity, a reduction of activity was observed when the residue Leu458 was substituted with the polar residue glutamine (Fig. 3e). However, no effect occurred when

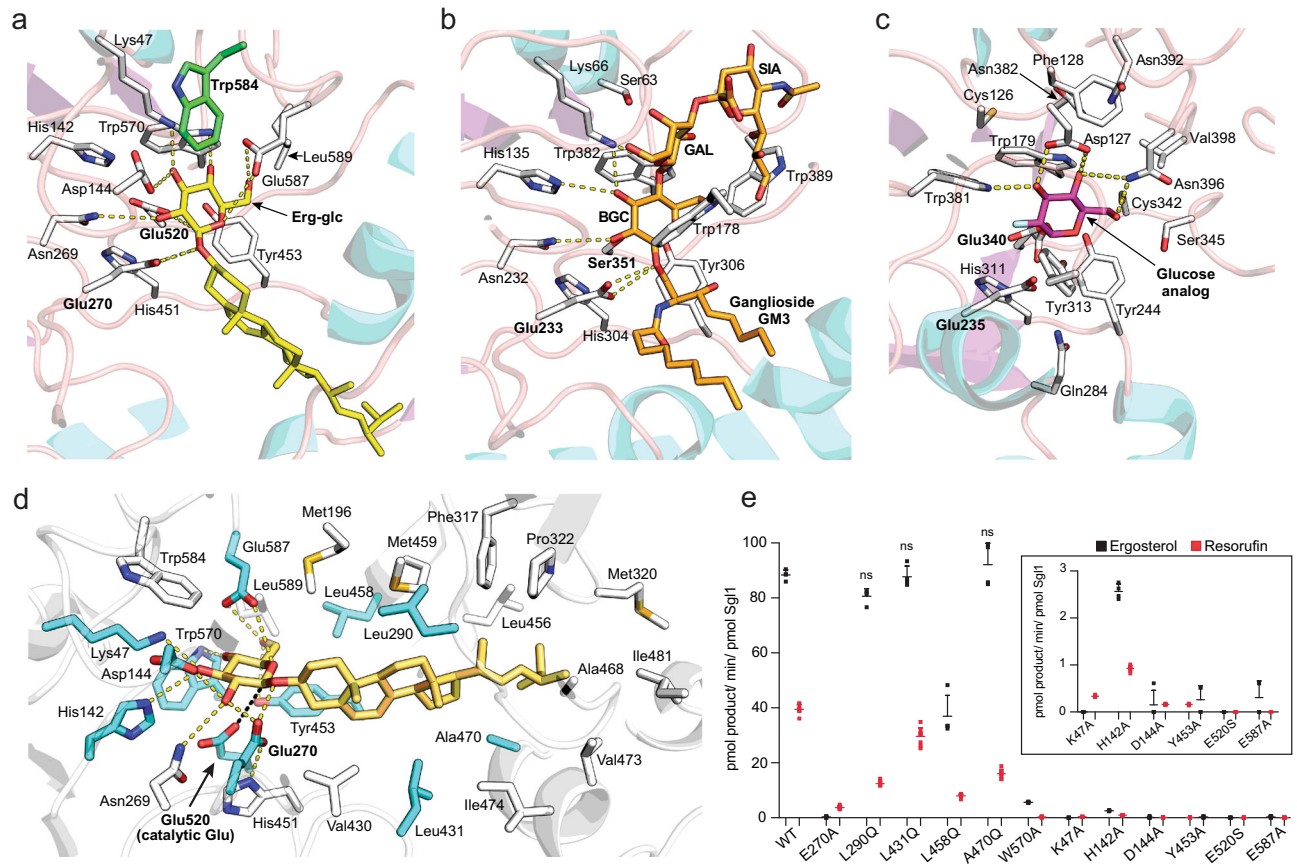

**Fig. 3 Structural basis for sterylglucoside specificity. a** Docking of erg-glc into Sgl1 shows the glucose in a similar position with EGCase II and Cerezyme, containing two catalytic glutamate residues, Glu270 and Glu520 (bold), found in the canonical topological positions of retaining β-glucosidases. Trp584 (green) provides steric hindrance to prevent the binding of multiple sugars and generate specificity for the single sugar of erg-glc. **b** EGCase II presented in the same view as Sgl1 highlighting the conserved glutamate residues Glu233 and Glu351 (bold). The point mutation E351S allowed co-crystallization of the protein–substrate complex. EGCase II has an unimpeded active site to hydrolyze gangliosides with multiple sugars. **c** Cerezyme co-crystallized with a glucose analog utilizes a distinct set of residues for glucose recognition, with a unique set of residues forming an active site cap for glucosylceramide specificity. **d** Docked erg-glc is recognized by several hydrogen bond contacts between residues Lys47, His142, Asn269, Glu270, Trp570, and Glu587 and the glucose moiety. The ergosterol moiety is surrounded by hydrophobic residues. Point mutants of green residues were generated to test their effects on erg-glc hydrolysis. **e** The point mutations of residues binding to the glucose moiety reduce or completely abolish Sgl1 activity towards the endogenous substrate erg-glc (black) or the artificial substrate res-glp (red) ($p < 0.001$). The inset graph diplays a zoomed in view of the residues with the lowest catalytic activities. Four residues around the ergosterol moiety were mutated to glutamine. L458Q reduces Sgl1 catalytic activity towards erg-glc and res-glp ($p < 0.001$), while L290Q, L431Q, and A470Q do not affect Sgl1 activity towards erg-glc ($p > 0.05$). Data represent the mean ± SD for $n = 4$ independent experiments with erg-glc and $n = 8$ with res-glp. Statistical analysis by one-way ANOVA, Dunnett's multiple comparison test. Source data are provided as a Source Data file.

Leu290, Leu431, or Ala470 were substituted with glutamine (Fig. 3e). Leu458 is close to the predicted glucose-binding site (Fig. 3d), which may explain the reduction of activity in comparison to the other residues that simply line the hydrophobic pocket. Together, this suggests the hydrophobic pocket of Sgl1 displays plasticity to accommodate the glutamine residues without loss of activity. Consistent with this, Leu431 and Ala470 are not universally conserved and are substituted with bulkier residues in some Sgl1 homologs (Supplementary Fig. 3).

To further investigate the structural mechanisms for Sgl1 substrate specificity, we used the same flexible ligand docking protocol to predict the binding modes of both synthetic C6-NBD-GlcCer and fungal GlcCer. C6-NBD-GlcCer bound within the Y-shaped pocket with the C6-acyl-NBD-group located in the buried hydrophobic cavity in a similar position to ergosterol and the sphingosine chain branching into the solvent exposed section of the Y-shaped cavity (Supplementary Fig. 5). The glucose moiety of C6-NBD-GlcCer was in an identical position to erg-glc, which is consistent with the low level of

activity against this synthetic substrate. In contrast, the binding pose of fungal GlcCer to Sgl1 was contorted, with the glucose moiety flipped into a non-canonical position that would not support hydrolysis (Supplementary Fig. 5). Taken together, our data suggest Sgl1 generates substrate specificity for erg-glc through its enclosed Y-shaped cavity, which limits hydrolysis of fungal GlcCer due to steric hindrance.

**Sgl1 inhibitors**. As *C. neoformans* Sgl1 is a promising therapeutic target for the treatment of fungal infections, we sought to identify small-molecule inhibitors of Sgl1 using a high-throughput screening (HTS) tiered-approach (Fig. 4a). The molecule res-glp was used as a substrate, as it has been successfully employed in HTS for other glucosidases[10,16]. After determining the $K_M$ and $k_{cat}$ for res-glp (Supplementary Fig. 6a–j), and optimization of the $Z'$-factor of our HTS assay (Fig. 4b), we screened 50,000 compounds to identify competitive inhibitors of Sgl1. A cocktail of 10 compounds per well were screened at 1 μM each and 35 hits with inhibition higher than 50% were selected for individual testing.

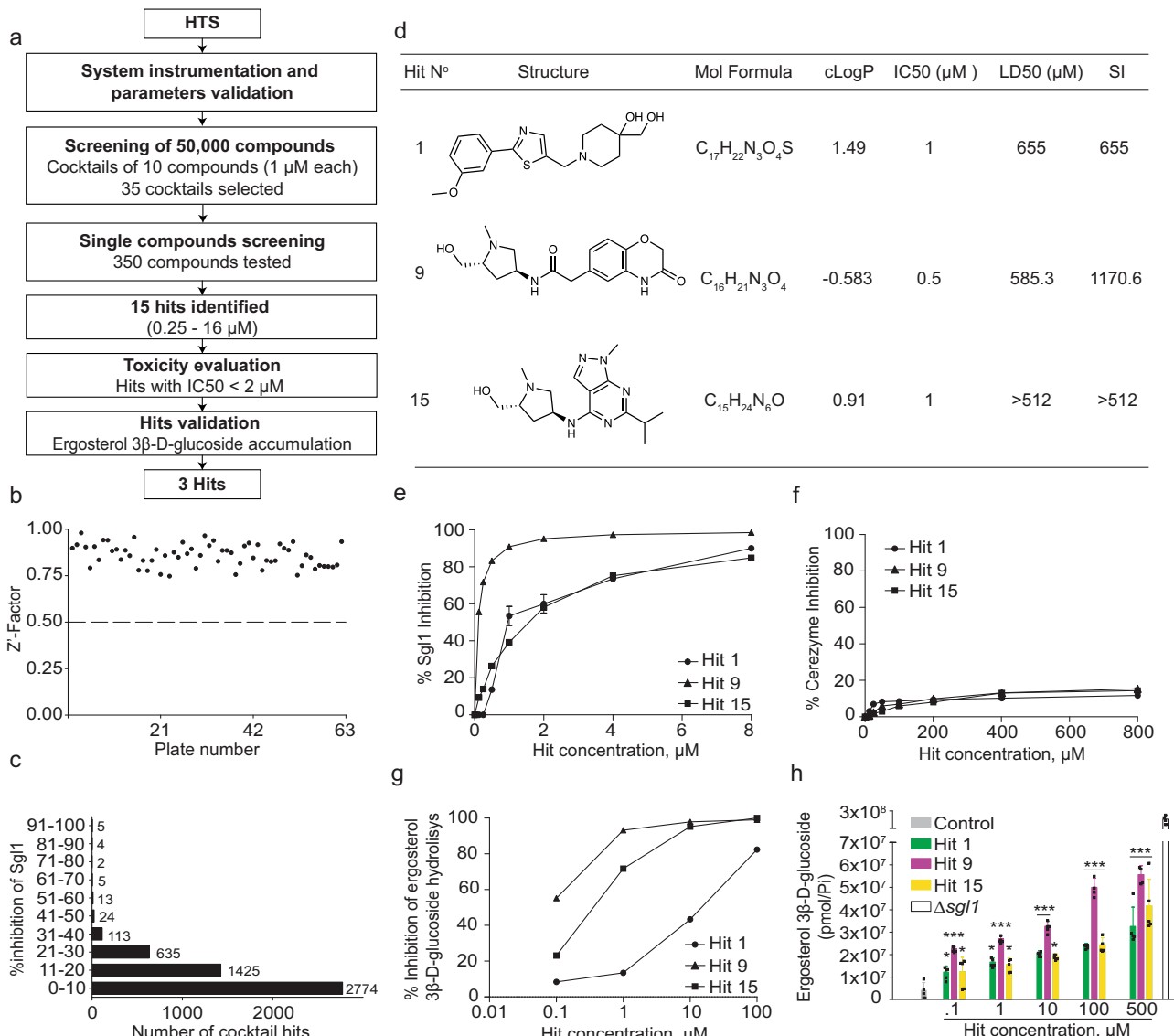

**Fig. 4 High-throughput small-molecule drug screening discovers three potent Sgl1 inhibitors with low toxicity. a** Flow chart of the HTS campaign. **b** Z'-factor graph of 63 cocktail plates. **c** Percentage of inhibition of all 5000 cocktail hits. **d** Top 3 hits selected for validation. Hits 1, 9, and 15 presented good solubility, $IC_{50}$ around 1 μM against Sgl1 and low toxicity to human cell line A549 (LD50 > 500 μM) resulting in a selectivity index (SI; SI = LD50/IC50)) higher than 500. **e** Dose–response curve of hits 1, 9, and 15 using resorufin-3β-D-glucopyranoside as substrate. Data represent the mean ± SD for $n = 2$ independent experiments. **f** Effect on Cerezyme activity using resorufin-3β-D-glucopyranoside as substrate for hits 1, 9, and 15. All three hits did not potently inhibit Cerezyme showing that these compounds are specific for Sgl1. Data represent the mean ± SD for $n = 2$ independent experiments. **g** Dose–response curves against Sgl1 for hits 1, 9, and 15 using the native substrate erg-glc. Data represent the mean ± SD for $n = 2$ independent experiments. **h** Erg-glc accumulation in *C. neoformans* (*Cn*) wild-type H99 after treatment with Hits 1 (green), 9 (magenta), and 15 (yellow) after 24 h of incubation. Erg-glc levels were quantified by LC-MS. All three hits significantly increased erg-glc levels compared to the control (untreated *Cn* H99 cells). Statistical analysis by one-way ANOVA, Dunnett's multiple comparison test; *$p < 0.05$, ***$p < 0.001$. Data represent the mean ± SD for $n = 5$ independent experiments. Control represents untreated wild-type *Cn* cells. At 500 μM the accumulation of erg-glc for Hits 1, 9, and 15 was about 10-fold less that the mutant *Cn* Δ*sgl1*. Source data are provided as a Source Data file.

From 350 single compounds tested, we identified 15 hits that were submitted for toxicity evaluation and validation of their ability to promote erg-glc accumulation in *C. neoformans* cells. This HTS campaign identified three hits (Hit 1, Hit 9, and Hit 15), belonging to two chemical classes that potently inhibited *Cn* Sgl1 activity (Fig. 4a–d). All three hits were re-synthesized via unambiguous synthetic routes (Supplementary Methods) with full characterization of their chemical structures by [1]H NMR, [13]C NMR, and high-resolution mass spectrometry. The chemical synthesis and characterization of Hits 1, 9, and 15 are provided in the Supplementary Information together with a LC-UV-MS analysis summary (Supplementary Fig. 7).

Hit 1 is a piperidine derivative with good solubility (clogP 1.49) (Fig. 4d) and an $IC_{50}$ of 1 μM in our primary screen with res-glp as the substrate (Fig. 4e). Hits 9 and 15 belong to the same chemical class with a common methyl-pyrrolidine ring followed by differing aromatic rings (Fig. 4d). Hits 9 and 15 have good solubility with $c$Log $P$ values of −0.583 and 0.91 (Fig. 4d), and $IC_{50}$'s of 0.5 and 1 μM against res-glp (Fig. 4e), respectively. All three hits were selective for Sgl1 and did not inhibit human Cerezyme either when using a synthetic res-glp substrate (Fig. 4f) or when using a physiological mammalian glucosylceramide substrate (Supplementary Fig. 8). Hits 1, 9, and 15 had low toxicity on the mammalian cell line A549 with lethal dose 50

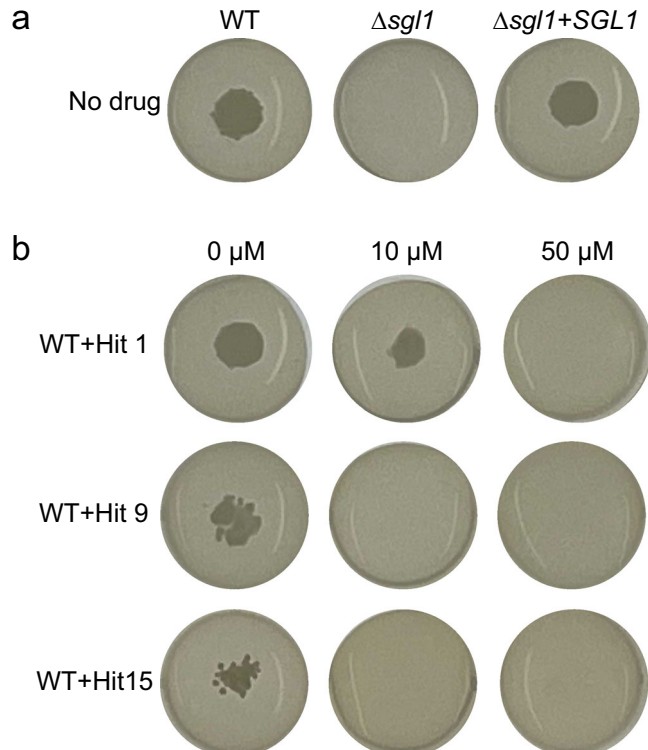

**Fig. 5 Effect of Sgl1 deletion or pharmacological inhibition on growth of C. neoformans. a** Deletion of Sgl1 gene in *C. neoformans* Δ*sgl1* affects growth in YNB agar containing 1% glucose in limited oxygen condition compared to wild type or reconstituted *C. neoformans* strains. **b** Treatment of *C. neoformans* WT with 50 μM Hit 1, 9, or 15 inhibit growth similarly to *C. neoformans* Δ*sgl1*.

($LD_{50}$) values of 655, 585.3, and > 512 μM (Fig. 4d), respectively. This results in selectivity indexes (SIs) higher than 500 (Fig. 4d).

To validate the selective inhibition of Sgl1 by these compounds, we first tested their ability to inhibit hydrolysis of erg-glc in vitro. Dose-dependent inhibition of erg-glc hydrolysis by Sgl1 was observed for all three compounds with $IC_{50}$ values in the low micromolar range (Fig. 4g).

**Erg-glc accumulation in *C. neoformans*.** Subsequently, we sought to validate the capacity of these Sgl1 inhibitors to promote the accumulation of erg-glc inside live *C. neoformans* cells. Upon treatment of *C. neoformans* wild-type H99 cells, we observed a significant and dose-dependent increase in the intracellular concentration of erg-glc, which was nearly undetectable in untreated cells (Fig. 4h). Maximal accumulation of erg-glc occurred at the highest concentration for each of the three compounds; however, erg-glc accumulated at the low micromolar concentrations that inhibit Sgl1 in vitro (Fig. 4g, h). Among the three hits, Hit 9 promoted higher accumulation of erg-glc in all concentrations tested. Hit 9 differs significantly from the control in all concentrations tested ($p < 0.001$) after 24 h of treatment. Similarly, Hits 1 and 15 present significant differences, especially in concentrations higher than 10 μM ($p < 0.001$) after 24 h of incubation. The time of exposure of *C. neoformans* cells to the compounds promotes similar accumulation of erg-glc between 3 and 24 h and begins decreasing after 48 h (Supplementary Fig. 9). Accumulation of erg-glc remained an order of magnitude less than the *C. neoformans* Δ*sgl1* strain, where Sgl1 is genetically deleted (Fig. 4h).

***C. neoformans* Δ*sgl1* exhibits a growth defect that is phenocopied by pharmacological inhibition of Sgl1.** Before proceeding to animal studies, we sought to develop a secondary assay to assess pharmacological inhibition of Sgl1 in *C. neoformans* cells. Previously, we and others failed to identify an obvious phenotype for the *C. neoformans* Δ*sgl1* strain[10,11] except for alterations to glucuronoxylomannan (GXM)[17], the main polysaccharide component of the *C. neoformans* capsule. Assessment of GXM composition is cumbersome and would require >100 mg of each hit compound. Thus, we re-tested the growth of *C. neoformans* Δsgl1 under conditions of nutrient and oxygen deprivation using yeast nitrogen base without amino acids, a glucose concentration of 1% and sealing the plates with clear polyolefin tape. Under these conditions, we identified a growth defect for the *C. neoformans* Δ*sgl1* strain (Fig. 5a). This phenotype was corrected when using the reconstituted strain (Δ*sgl1* + *SGL1*), in which wild-type *C. neoformans* SGL1 gene is re-introduced and expressed using its own promoter into the Δ*sgl1* strain (Fig. 5a). The identification of this in vitro growth defect phenotype controlled by Sgl1 is important because it does allow the screening of anti-Sgl1 compounds.

In fact, having established a secondary cell-based assay, we next assessed if pharmacological inhibition of Sgl1 phenocopied the growth defect of *C. neoformans* Δ*sgl1*. Treatment with a concentration of 50 μM of either Hit 1, 9, or 15 resulted in a lack of growth of wild-type *C. neoformans* (Fig. 5b). Consistent with their higher potency in vitro and more robust accumulation of erg-glc, Hit 9 and 15 also abrogated *C. neoformans* growth at the lower concentration of 10 μM. We concluded that pharmacological inhibition of Sgl1 can mimic the effect of genetic deletion of Sgl1.

**Hit 1 prevents brain dissemination of *C. neoformans* in a mouse model of infection.** Encouraged by these promising results, we assessed the therapeutic potential of Sgl1 inhibition in a well-characterized mouse model of *C. neoformans* infection. Despite its lower potency in vitro and in cells, Hit 1 was chosen for animal experiments as it was predicted to have more optimal pharmacokinetic properties. In an initial experiment, female CBA/J mice were infected intranasally with *C. neoformans* H99 cells and treated daily through intraperitoneal administration with 10 mg/kg/day of Hit 1 or vehicle. Hit 1 did not reduce lung colony-forming units (CFUs) compared to vehicle-treated mice (Fig. 6a). However, no CFU were recovered from the brain 14 days after infection in mice receiving Hit 1 (Fig. 6b).

Intrigued by the ability of Hit 1 to prevent brain dissemination, we reasoned Sgl1 inhibition in combination with a current standard of care treatment may prove more effective in controlling *C. neoformans* infection. In an independent experiment, female CBA/J mice were inoculated with *C. neoformans* H99 cells, and subsequently treated with Hit 1 administered intranasally, fluconazole via intraperitoneal injection, or Hit 1 in combination with fluconazole. After 14 days of treatment, fluconazole alone slightly reduced lung CFU ($p < 0.05$); however, the addition of Hit 1 further enhanced this activity ($p < 0.01$, Fig. 6c) compared to either treatment alone. Fluconazole alone did not prevent *C. neoformans* brain dissemination compared to vehicle treatment. In contrast, Hit 1 alone or in combination with fluconazole resulted in no CFU recovered from the brain (Fig. 6d). Overall, Sgl1 inhibition displays efficacy in controlling *C. neoformans* infection in vivo, thus validating Sgl1 as a potential broad-spectrum antifungal target.

**Sgl1-inhibitor complex structures.** The initial set of Sgl1 inhibitors disclosed here have not undergone optimization. To aid

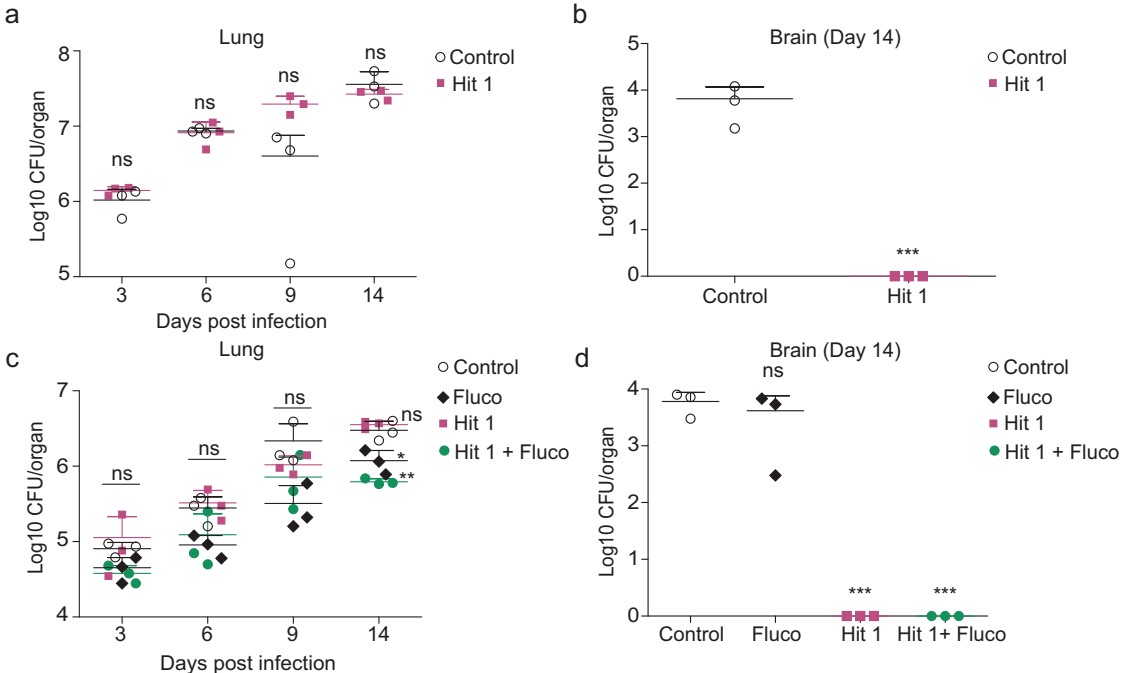

**Fig. 6 Hit 1 prevents brain dissemination in a mouse model of cryptococcosis. a** Female CBA/J mice were inoculated with $1 \times 10^6$ *Cryptococcus neoformans* H99 cells in log phase and the treatment started at the same day of infection via intraperitoneal (IP) with 10 mg/kg/day of hit 1. The CFU was recovered from the lungs at days 3, 6, 9, and 14 after infection and there was no significant difference among untreated and treated mice. $n = 3$ mice for each group. Statistical analysis by one-way ANOVA, Tukey's multiple comparison test. $^{ns}p > 0.05$. **b** No yeast cells were recovered from the brain after 14 days of infection from mice treated with hit 1. $n = 3$ mice for each group. Statistical analysis by unpaired Student's *t*-test. $^{***}p < 0.001$. **c** Mice were inoculated with $5 \times 10^5$ H99 cells and hit 1 was administered intranasally (IN) at 10 mg/kg/day alone and in combination with fluconazole (IP) at the same dose. There was a significant difference among CFU recovered from the lungs of animals treated with fluconazole alone ($p < 0.05$) and in combination with hit 1 ($p < 0.01$). $n = 3$ mice for each group. Statistical analysis by one-way ANOVA, Tukey's multiple comparison test. $^{ns}p > 0.05$; $^*p < 0.05$, $^{**}p < 0.01$. **d** There was no brain dissemination of yeast cells when animals received treatment with hit 1 alone or in combination with fluconazole ($p < 0.001$); however, fluconazole administered alone results in a similar fungal burden in brain compared to vehicle-treated mice (control) ($p > 0.05$). $n = 3$ mice for each group. Statistical analysis by one-way ANOVA, Tukey's multiple comparison test, $^{ns}p > 0.05$; $^{***}p < 0.001$. Source data are provided as a Source Data file.

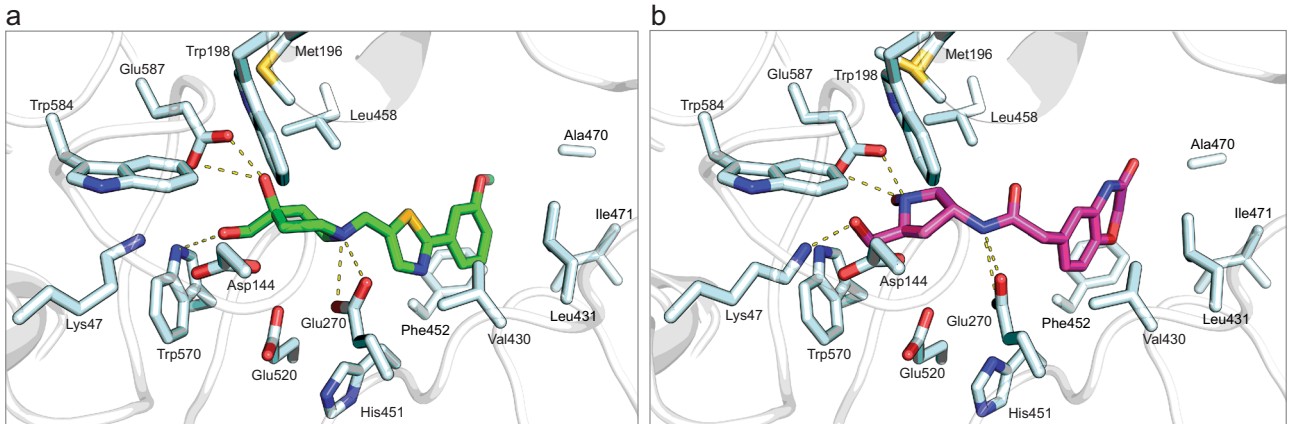

**Fig. 7 Hit 1 and Hit 9 bind within the active site of *Cryptococcus neoformans (Cn)* Sgl1. a** Co-crystal structures of Hit 1 (green) and **b** Hit 9 (magenta) in the active site of *Cn* Sgl1. Hit 1 makes hydrogen bond contacts to Glu270 and Trp570. Hit 9 makes hydrogen bond contacts with Lys47, Glu270, and Glu587.

future medicinal chemistry optimization, we determined co-crystal structures of Hit 1 and Hit 9 with *C. neoformans* Sgl1 at 2.8 and 2.3 Å resolutions, respectively (Fig. 7 and Table 1). Hit 15 has not yet been successfully co-crystallized. Strong electron density for Hit 1 and Hit 9 was observed in all four Sgl1 molecules in the asymmetric unit, which allowed unambiguous modeling (Supplementary Fig. 10a, b). Both compounds bound within the active site and formed a series of polar interactions near the

catalytic site with the planar aromatic rings of the compounds occupying space in the buried hydrophobic pocket (Fig. 5a, b). The overall position of Hit 1 and Hit 9 were notably similar to the predicted docked erg-glc-binding pose. The piperidine ring of Hit 1 formed hydrogen bond interactions with the active site residues Glu270, Trp570, and Glu587, while the methyl-pyrrolidine ring of Hit 9 formed key interactions with Lys47, Glu270, and Glu587. With the exception of the catalytic residue Glu270, the residues

that directly interact with Hit 1 and 9 are not conserved with human Cerezyme, which is consistent with specific inhibition of Sgl1.

## Discussion

Here we confirm *Cn* Sgl1 as the first sterylglucosidase to be functionally characterized, reveal the structural basis for erg-glc substrate specificity, and identify first-in-class inhibitors of Sgl1 that cause accumulation of erg-glc in cells, phenocopy a growth defect in the *Cn Δsgl1* mutant strain, and are efficacious in a mouse model of *Cn* infection. These compounds represent the first chemical tools to manipulate erg-glc levels in any species and provide an important starting point for medicinal chemistry optimization.

The mechanism of glycolipid discrimination by Sgl1 is straightforward. Sgl1 contains an enclosed and sterically restrained Y-shaped pocket that can bind erg-glc but cannot accommodate glucosylceramide for productive catalysis. The residues that define the Y-shaped pocket are largely conserved in other Sgl1 fungal homologs (Supplementary Fig. 3) but not present in human[13] or bacterial[12] endo-glucoceramidases. This allows the identification of Sgl1 homologs in other pathogenic fungi (Supplementary Fig. 3), including *Aspergillus* and *Candida*, and suggests that pharmacological inhibition of Sgl1 represents a relevant strategy to treat fungal infections.

Sgl1 is an ideal pharmacological target with broad spectrum since Sgl1 is widely conserved in fungi and there is no obvious homolog in humans. Based on previous studies, it became apparent that Sgl1 is a key virulence factor in *C. neoformans* that modulates fungal pathogenesis[11,17]. Inactivation of Sgl1 in the *C. neoformans Δsgl1* mutant not only improves the outcome of infection, notably without any fungicidal activity, but also enables the host with protective immunity against wild-type *C. neoformans* even when mice are CD4+ T cell depleted[11]. This is especially relevant as most fungal infections, including cryptococcosis, aspergillosis, and candidiasis, occur in immunocompromised patients. Sgl1 inhibitors could be applicable as a preventive therapy in patients more susceptible to develop fungal diseases or/and as immunoadjuvants administered in combination with regular antifungal therapy. Consequently, by inhibiting Sgl1 pharmacologically, it would be possible not only to help the host to eliminate the fungus but potentially also to induce protection against a relapse or a secondary infection. Our finding that pharmacological inhibition of Sgl1 prevents brain dissemination of wild-type *C. neoformans* suggests this approach has promise. Optimization of inhibitor potency and pharmacological properties may enhance these effects to also yield clearance from the lung.

Improved potency is likely to be key to driving accumulation of erg-glc to the levels seen in the *C. neoformans Δsgl1* mutant strain, which may be a key factor in improving lung clearance in vivo. Alterations of the piperidine and methyl-pyrrolidine rings of Hit 1 and Hit 9/15 to modify the chemical interactions in the glucose-binding site represent one strategy to drive improved potency. Another strategy would involve expansion of the inhibitors to occupy both arms of the Y-shaped pocket. In view of all this, the crystal structure of Sgl1 with two specific inhibitors now enables rational drug development against this therapeutic target for fungal infections.

There are reports of piperidine-containing iminosugars that inhibit α-galactosidase and α-mannosidase[18]. Derivatives of thiazole, which is a moiety found in Hit 1, have also been reported to inhibit α-glucosidase[19,20]. However, to our knowledge, there are no inhibitors with similar chemical structures to the lead compounds reported here that target other β-glucosidases or that have

been reported as antifungal agents. Thus, optimization of these first-in-class Sgl1 inhibitors is now urgent and improved potency will also help specify if other glucosidases in *C. neoformans* cells are inhibited.

The newly discovered growth defect of *C. neoformans Δsgl1* under conditions of nutrient deprivation and low oxygen will aid therapeutic development by providing a simple readout of inhibitor efficacy to precede more in-depth animal studies. This growth defect is intriguing as the low oxygen is selectively present in conditions that more closely mimic the host environment during the infection process. When the *Δsgl1* reaches the alveolar space, the alveolar partial pressure of $O_2$ (PAO$_2$) is high (~100 mmHg) and under this condition the mutant is able to survive. But when *Δsgl1* in phagocytosed, it is now exposed to a very low (1–10 mmHg) intracellular $O_2$ (iPO$_2$)[21]. Similarly, when *Δsgl1* moves into the interstitial space, the tissue partial pressure of oxygen (PtO$_2$) is much lower than the PAO$_2$, particularly within the lung parenchyma and during the granulomatous inflammatory response (4–30 mmHg)[22–24]. Most likely, by decreasing Sgl1 activity Hit 1 is able to control wild-type *C. neoformans* in this organ, as this activity is required for fungal replication in low oxygen. Thus, Hit 1 may prevent the fungus from leaving the lung. On the other hand, the brain is an organ in which oxygen concentration is low, ranging from 11.4 to 53.2 mmHg[25]. So, it is possible that treatment with Hit 1 also inhibits fungal replication in this organ. Interestingly, ergosterol biosynthesis is required for *C. neoformans* replication and survival at low oxygen[26–28]. Thus, it is possible that by releasing ergosterol, Sgl1 contributes to promote fungal survival and replication in oxygen-limiting environments. These observations suggest that targeting Sgl1 with a pharmacological inhibitor may be highly effective in protecting against cryptococcosis and potentially against other fungal infections, as Sgl1 is present in other pathogenic fungi.

## Methods

**Chemical compounds.** A 50,000 compounds DIVERSet-CL library and individual hits were purchased from ChemBridge (San Diego, CA). Hits 1, 9, and 15 were re-synthesized by the laboratory of Dr. Iwao Ojima at Stony Brook University (Supplementary Methods and Supplementary Fig. 7).

**Plasmid.** *C. neoformans* Sgl1 wild-type, Δ723–777, Δ609–634, Δ836–851, and 12 point mutations were gene synthesized, codon optimized for expression in *E. coli* and inserted into a ppSUMO plasmid by BioBasic (Canada).

**C. neoformans Sgl1 overexpression and purification.** *C. neoformans* Sgl1 plasmids were transformed into BL21 (DE3) RIPL cells (Agilent Technologies) for protein overexpression. Cells were grown at 37 °C in Terrific Broth to an OD$_{600}$ of 1.5 and then were cooled at 10 °C for 2 h. After, the protein expression was induced with 100 μM isopropyl β-D-1-thiogalactopyranoside (IPTG) at 15 °C overnight before harvesting. Cell pellets from 1 L culture were resuspended in a lysis buffer comprised of 50 mM Tris pH 7.5, 500 mM NaCl, 60 mM imidazole, 5% glycerol, 1% Triton X-100, and 2 mM β-mercaptoethanol (βME), and then lysed by sonication at 85 amplitude with cycles of 2 s during 1.5 min. This procedure was repeated five times and the resulting cell lysates were centrifuged at 48,380 × *g*.

Thereafter, *C. neoformans* Sgl1 was purified using a HisTrap FF column and eluted with an increased imidazole concentration of 300 mM in 30 mM Bis-Tris pH 6 buffer with 500 mM NaCl, 5% glycerol, and 2 mM βME. Resulting fractions were supplemented with 8 mM βME and 10 mM dithiothreitol (DTT) and then incubated overnight at 4 °C with purified ULP-1 for His-SUMO tag cleavage. Finally, the protein was applied to a Superdex 26/60 HiLoad 200 column (GE Healthcare) equilibrated with 50 mM Bis-Tris (pH 6), 150 mM NaCl, 10 mM βME, and 2 mM DTT. Purified protein was concentrated to 10 mg/mL, flash-frozen, and stored at −80 °C.

**Mapping disordered regions of *C. neoformans* Sgl1 through H/D exchange mass spectrometry (HDX-MS).** HDX-MS reactions were performed in a similar manner as is outlined in previous publications[29]. HDX reactions were conducted in a final reaction volume of 50 μL with a final Sgl1 concentration of 0.6 μM (30 pmol). Exchange was carried out in triplicate for a single time point (3 s at 4 °C). Hydrogen deuterium exchange was initiated by the addition of 48 μL of D$_2$O

buffer (100 mM NaCl (20 mM pH 6.0 Bis-Tris), 90.5% D$_2$O (V/V) to 1 µL of protein and 1 µL of H$_2$O. Exchange was terminated by the addition of ice-cold acidic quench buffer, resulting in a final concentration of 0.6 M guanidine-HCl and 0.9% FA post quench. Samples were immediately frozen in liquid nitrogen and stored at −80 °C. Fully deuterated samples were generated by first denaturing the protein in 2 M guanidine for 5 min at 20 °C. Following denaturing, 48 µL of D$_2$O buffer was added to the denatured protein and allowed to incubate for 5 min at 20 °C before quenching. Samples were flash frozen and stored at −80 °C until injection onto the ultra-performance liquid chromatography (UPLC) system for proteolytic cleavage, peptide separation, and injection onto a QTOF for mass analysis.

Protein sample was rapidly thawed and injected onto a UPLC system kept in a cold box at 2 °C. The protein was run over two immobilized pepsin columns (Applied Biosystems; Porosyme 2-3131-00) and the peptides were collected onto a VanGuard Precolumn trap (Waters). The trap was eluted in line with an ACQUITY 1.7 µm particle, 100 × 1 mm$^2$ C18 UPLC column (Waters), using a gradient of 5–36% B (Buffer A 0.1% formic acid, Buffer B 100% acetonitrile) over 16 min. MS experiments were performed on an Impact QTOF (Bruker) and peptide identification was done by running tandem MS (MS/MS) experiments run in data-dependent acquisition mode. The resulting MS/MS dataset was analyzed using PEAKS7 (PEAKS) and a false discovery rate was set at 1% using a database of purified proteins and known contaminants[30]. HDExaminer Software (Sierra Analytics) was used to automatically calculate the level of deuterium incorporation into each peptide. All peptides were manually inspected for correct charge state and presence of overlapping peptides. Deuteration levels were calculated using the centroid of the experimental isotope clusters, with correction for back exchange generated using the fully deuterated sample. The mass spectrometry proteomics data have been deposited to the ProteomeXchange Consortium via the PRIDE partner repository[31] with the dataset identifier PXD024981.

**Se-Met Sgl1 for phasing.** *C. neoformans* Sgl1 Δ723–777 selenomethionine-enriched (Se-Met) was generated by transforming the Δ723–777 plasmid into BL21 (DE3) RIPL cells. The expression of Se-Met substituted proteins in non-methionine auxotrophic *E. coli* was performed according with Van Duyne et al.[32], where a feedback inhibition of methionine biosynthesis is facilitated by adding amino acids prior to induction.

Initially, a pre-culture in Luria broth (LB) was added (1:100) to 1 L of M9 minimal medium supplemented with 1 mL of vitamins (1000× solution: 0.5 g riboflavin, 0.5 g niacinamide, 0.5 g pyridoxine monohydrate, and 0.5 g thiamine per 500 mL volume) and incubated at 37 °C. After reaching an OD$_{600}$ of 1.5, a feedback inhibition amino acids mix was added as solids. The solid mixture (lysine 0.5 g, threonine 0.5 g, phenylalanine 0.5 g, leucine 0.25 g, isoleucine 0.25 g, valine 0.25 g, L(+)selenomethionine 0.25 g) was mixed very thoroughly with a spatula and divided into five portions (0.5 g each per 1 L culture) and the amino acids were added to the culture flasks. After 15 min, the expression was induced with 100 µM IPTG and the culture was incubated at 15 °C overnight before harvesting.

**Crystallization and data collection.** The crystallization process was performed using hanging drop vapor diffusion with a well solution of 10% polyethylene glycol (PEG) 8000, 0.2 M MgCl$_2$, and 0.1 M Tris pH 7.5 at room temperature. In all, 1.5 µL of *C. neoformans* Sgl1 Δ723–777 at 5 mg/mL was added to an equal volume of the well solution. Micro-seeds from previous crystal drops at the same condition were diluted 1:10 and 0.2 µL was added per drop. The crystals obtained were frozen in a cryoprotectant solution containing the same components of the drop solution plus 40% glycerol. Co-crystals with inhibitors were obtained using the same condition with hit 1 at 5.6 mM or hit 9 at 2.8 mM and the same protein concentration.

Native diffraction data were collected at Brookhaven National Lab NSLS AMX beamline 17-ID-1, SAD diffraction data with Se-Met protein and native diffraction data with hit 1 were collected at the FMX beamline 17-ID-2. SAD diffraction data were collected in 15 degree wedges. Hit 9 diffraction data were collected at the Advanced Photon Source GM CAT 23ID-B beamline at Argonne National Lab. All data were processed using xia2 DIALS in CCP4 (refs. [33–35]).

**Structure determination and refinement.** *C. neoformans* Sgl1 phasing procedure was carried out in Phenix using Autosol with 18 Se-Met sites identified, and an initial model generated by Autobuild[36–38]. Posteriorly, the data were submitted to manual modeling in coot and several refinements in Phenix. After, the generated model was used as a search model in Phaser for molecular replacement with the 2.13 Å native *C. neoformans* Sgl1 Δ723–777 (refs. [39–41]). Additional model building in Coot and refinement in Phenix produced the final model (Table 1, PDB code: 7LPO). This model was used for molecular replacement for the 2.85 Å hit1 (PDB code: 7LPP) and 2.32 Å hit 9 (PDB code: 7LPQ) co-crystal data sets. Polder map was used to improve the electron density of hit 1.

**Docking.** The docking study on the lipid substrates of Sgl1 was carried out using DOCK6 (ref. [42]) and its fixed anchor docking (FAD) algorithm[43]. Protein coordinates were obtained through crystallographic data of Sgl1 in complex with hit 9. The reference ligand coordinates were deleted, and the protein was processed

through the DockPrep module of UCSF Chimera[44] to add hydrogens and partial atom charges under the AMBER FF14SB[45] protein parameters. Docking spheres were generated over the surface contours of the protein and those within 8 Å of the reference ligand were retained for docking anchor placement. The docking grid was calculated using an extended length box with a 20 Å cutoff from the hit 9 ligand to account for the considerably larger size of these lipid substrates with a grid point spacing of 0.4 Å. The electrostatic interactions were modeled with a distance-dependent dielectric of 4r and a 6–9 Lennard–Jones potential for van der Waals interactions.

Lipid substrate structures for docking were constructed in ChemDraw starting from two-dimensional structures and converted into three-dimensional coordinates using the Avogadro[46] molecular editor and the OpenBabel[47] toolkit parameters. The structures were then minimized under the MMFF94 (refs. [48–52]) forcefield with the lowest energy chair conformations of glucose and non-aromatic steroid rings used whenever applicable.

Docking followed a three-stage protocol. First, flexible (FLX) docking[43] of ergosterol 3β-D-glucoside was conducted utilizing all retained docking spheres where a reasonable position of the bound glucosyl appendage was found. Second, the glucose coordinates of the two other lipids C6-NBD-glucosylceramide and fungal glucosylceramide were superimposed on the coordinates of the docked erg-glc using the Match function in UCSF chimera. All docking spheres outside of the glucose-binding site were then deleted. Third, the two glucosides were docked using the superimposed glucose coordinates as a "fixed anchor" under the FAD algorithm.

**Kinetic evaluation of ergosterol 3β-D-glucoside and C6-NBD-glucosylceramide.** A 50 µL volume of *Cn* Sgl1 wild type and 50 µL of mixed micelles of lipid and Triton X-100 in 50 mM Bis-Tris buffer with 150 mM NaCl, 5 mM βME, and 5 mM DTT were mixed and incubated at 37 °C for 10 min (erg-glc, Avanti Polar Lipids) and 1 h (C6-NBD-glcCer, Matreya). For the substrate erg-glc the amount of protein was 5 ng. 150 ng protein was used for reactions with C6-NBD-GlcCer. After the incubation period, each reaction was quenched with a 2:1 chloroform:methanol solution and the organic phase was collected and dried. Then, the lipid content was resuspended in 50 µL methanol and applied to HPLC (Agilent Technologies). Total erg-glc and ergosterol was detected using absorbance at 282 nm on a C-8 column with a flow rate of 0.5 ml/min in methanol/water 90:10 ratio buffered with 1 mM ammonium formate and 0.2% formic acid. A 85:15 ratio of the same eluent was used for C6-NBD-GlcCer and C6-NBD-ceramide monitoring the fluorescence at 470/530 nm excitation/emission. The total area of product was normalized by the total area of its respective substrate.

**Enzymatic evaluation of *C. neoformans* Sgl1 against the fungal glucosylceramide.** To verify the enzymatic activity of *C. neoformans* Sgl1 against the fungal long-chain glucosylceramide (GlcCer, OH-Δ8,9-methyl-glucosylceramide), kindly provided by Khojin (Japan). The reactions used 10 µg of substrate and 10 or 50 µg of *C. neoformans* Sgl1 or cerezyme (Genzyme Corporation, Cambridge, MA, USA) in 100 µL acetate buffer pH 5.5 incubated at 37 °C for 4 h[11]. The reaction was terminated by adding 300 µL of chloroform/methanol (1:1 ratio) and the lower phase was collected and dried. 9-Methyl-ceramide production was monitored by liquid chromatography-mass spectrometry (LC-MS).

**Resorufin-3β-D-glucopyranoside enzymatic assay.** Resorufin-3β-D-glucopyranoside (res-glp, Sigma-Aldrich) enzyme kinetic assays were performed according to methods described by Urban et al.[16] with modifications. Reaction buffer contained 50 mM citric acid, 176 mM K$_2$HPO$_4$, pH 6, 0.01% Tween-20, and 10 mM sodium taurocholate. The reactions were carried out in black 96-well plates using 20 ng of enzyme with 100 µM res-glp substrate. In all, 10 µL/well of res-glp substrate was added to each well and the reaction was initiated by adding 20 µL enzyme (20 ng). After incubation at 37 °C for 30–40 min, the fluorescence was read at an excitation of 570 (±10) nm and an emission of 610 (±10) nm on a VersaMax™ Microplate Reader from Molecular Devices. A standard curve was prepared with serial dilutions of the free fluorophore, resorufin, in the same volume of assay buffer. All reactions were linear with respect to time and protein concentration and the pH, temperature, and dimethyl sulfoxide (DMSO) optimal tolerance was verified.

**High-throughput screening (HTS) for Sgl1 inhibitors.** A ChemBridge DIVERSet-CL library (San Diego, CA) containing 50,000 compounds was screened to identify small molecules that inhibit *C. neoformans* Sgl1. The library was prepared in a 96-well plate format containing a cocktail of 10 compounds per well at 1 mM each in 100% DMSO. The cocktail plates were first diluted to 100 µM each (1:10 dilution in Dulbecco's phosphate-buffered saline, DPBS) resulting in 10% DMSO. In order to screen the compound cocktails at 1 µM in 1% DMSO in a final reaction volume of 30 µL an aliquot of 3 µL was added to 17 µL of buffer containing 20 ng of enzyme and the reaction was started adding 10 µL of 100 µM res-glp. A negative control without enzyme and positive controls with and without DMSO were prepared. The plates were incubated at 37 °C for 40 min.

The Z'-factor was used for the assessment of the efficiency of the HTS assay for each individual plate. Values greater than 0.5 has been considered as an indicator of

an excellent screening assay quality[53]. Compound cocktails from plates with $Z'$-factor higher than 0.5 and showing > 50% inhibition compared to the control well (1% DMSO but no drug) were selected for tests with each individual compound in serial dilution from 0.125 to 16 μM. The top three hits identified were re-tested in serial dilution against Sgl1 and cerezyme.

**Toxicity evaluation of hits.** The single compounds with $IC_{50}$ under 2 μM were selected to toxicity evaluation against the human lung epithelial cancer cell lines A549. The cells were maintained in Dulbecco's modified eagle medium containing 10% fetal bovine serum and 1% penicillin–streptomycin. At passage 11, $10^5$ cells were transferred into each well of 96-well plates and cultured for 18–24 h for the cells to adhere to the well surface. After, the medium was removed and a fresh medium containing the selected compounds at a concentration range of 1–512 μM were added to the wells. Controls with equivalent serial concentrations of DMSO were also evaluated and compared with a control without DMSO. The plate was incubated at 37 °C with 5% $CO_2$. After 24 h, the supernatant was removed and 50 μL of 5 mg/mL 3-(4,5-dimethylthiazol-2-yl)-2,5-diphenyltetrazoliumbromide (MTT) solution in phosphate-buffered saline (PBS) was added to each well. The plates were incubated for an additional 4 h and the formazan crystals formed inside the cell was dissolved by adding 50 μL DMSO. Hence, the absorbance was measured at 570 nm and the calculated $LD_{50}$ was divided by the $IC_{50}$ of each hit in order to determine the SI of each compound. Compounds with SI value higher than 500 were considered for further validations.

**Ergosteryl 3β-D-glucoside accumulation.** C. neoformans H99 was cultivated in yeast nitrogen base (YNB) broth for 24 h at 37 °C with shaking. A pellet with $5 \times 10^8$ cells was treated during 24 h with all the hits selected, according to the toxicity criteria, in various concentrations. A previous minimal inhibitory concentration assay was performed in accordance with the guidelines in the CLSI document M27-A3 (ref. [54]), to determine the minimal inhibitory concentration and selecting a concentration range that did not affect the yeast growth. After that, the resultant pellets were re-counted and used for lipid extraction. Thenceforth the total lipid was extracted as described previously by Singh et al.[55]. The dried samples were resuspended in chloroform/methanol 2:1 ratio for LC-MS analysis. A standard erg-glc from Avanti Polar Lipids was used as a control for the calibration curve. Data were normalized to the total inorganic phosphate content in the sample.

**Growth inhibition assays.** Five microliters containing 100 cells of C. neoformans H99 were inoculated in solid YNB media without amino acids plus 1% w/v glucose. Plates were then sealed with clear polyolefin sealing tape (ThermoFisher) and incubated at 37 °C for 72 h. Hits 1, 9, and 15 were added to the media at concentrations of 10 and 50 μM. H99, Δsgl1, and Δsgl1 + SGL1 strains were used as controls without compounds.

**In vivo assays.** For animal studies, 4-week-old CBA/J (Envigo) female mice were used. Mouse experiments were performed in full compliance with a protocol approved by Stony Brook University (IACUC number 341588) and in compliance with the United States Animal Welfare Act (Public Law 98–198). The experiments were carried out in facilities accredited by the Association for Assessment and Accreditation of Laboratory Animal Care. Mice were divided into sets of 12 mice for each treatment or control group (4 animals per cage) kept in an animal room (temperature, 22–24 °C; relative humidity, 35–45%) with artificial 12–12 h dark/light cycle. For the experiment with intraperitoneal (i.p.) treatment, mice were infected intranasally with 10 μL of a suspension containing $1 \times 10^6$ C. neoformans H99 cells and subsequently treated through i.p. injection with 10 mg/kg/day of hit 1 in a final volume of 100 μL of vehicle containing 10% PEG 200 in PBS. The untreated animals received 100 μL of vehicle. For the experiment with intranasal (i.n.) treatment, mice were infected with $1 \times 10^5$ C. neoformans H99 cells and received i.n. treatment with hit 1 in a final volume of 10 μL. Another group received treatment with 10 mg/kg/day of fluconazole i.p. and a third group received the combined treatment of hit1 by i.n. and fluconazole by i.p. Animals receiving only vehicle i.n. were used as a control group. In both experiments, three mice of each group were euthanized at day 3, 6, 9, and 14 post infection for CFU recovery from lungs and CFU recovery from brain on day 14.

**Statistical analysis.** Statistical analysis was performed using Prism 5 (GraphPad Software) and conducted on data from three or more biologically independent experimental replicates. Error bars displayed on column graphs represent the mean ± SD of at least three independent experiments and individual data points are plotted. Statistical significance was analyzed using unpaired Student's $t$-test for two groups or ordinary one-way ANOVA with Dunnett multiple comparison test and one-way ANOVA with Tukey's multiple comparison test for multiple groups, with $*p < 0.05$, $**p < 0.01$, and $***p < 0.001$ considered significant.

**Reporting summary.** Further information on research design is available in the Nature Research Reporting Summary linked to this article.

## Data availability

Coordinates and structure factors have been deposited in the Protein Data Bank under accession codes 7LPO, 7LPP, 7LPQ. The mass spectrometry proteomics data have been deposited to the ProteomeXchange Consortium via the PRIDE partner repository[31] with the dataset identifier PXD024981. Pdb files of erg-glc, C6-NBD-glcCer, and fungal glcCer docked with C.neoformans Sgl1 are included with this manuscript as Supplementary Data 1 file. EGCase II and cerezyme structures displayed in Fig. 2d and e can be found under the accession codes 2OSX and 6TJQ, respectively. Source data are provided with this paper.

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

## Acknowledgements

We thank the staff at the AMX and FMX (NSLS-II) and GM-CAT (APS) beamlines for assistance during data collection, Yong Mi Choi (Airola lab) for help with Sgl1 purification, and Robert Rieger and Izolda Mileva from Stony Brook Proteomics and Lipidomics facilities who provided valuable assistance for LC-MS analysis. We are also grateful to Dr. Tadashi Honda, Discovery Chemistry Laboratory, for his valuable advice on the chemical synthesis of Hits 1, 9, 15, as well as Dr. Bela Ruzsicska, Analytical Instrumentation Laboratory, for HRMS analysis at the Institute of Chemical Biology and Drug Discovery, Stony Brook University. This work was supported by a grant from the Feldstein Medical Foundation 1169048 (M.V.A.), a Stony Brook School of Medicine Fusion Award 1149556 (M.V.A. and M.D.P.), National Institutes of Health grants R35GM128666 (M.V.A.), AI136934 (M.D.P.), AI116420 (M.D.P.), AI125770 (M.D.P.), and R35GM126906 (R.C.R.), a Merit Review Grant I01BX002924 from the Veterans Affairs Program (M.D.P.), the NSERC Discovery Grant NSERC-2020-04241 (J.E.B.), and the Michael Smith Foundation for Health Research (J.E.B., Scholar Award 17686). Maurizio Del Poeta is a Burroughs Welcome Investigator in Infectious Diseases.

## Author contributions

N.P.S. performed all experiments with *C. neoformans* Sgl1 including protein purifications, crystallization experiments, H.T.S. and hit validation, growth assays, and animal experiments. N.P.S. and M.V.A. determined and refined the final crystal structures. R.M.H. performed the HDX-MS experiments for Sgl1. R.M.H. and J.E.B. analyzed all HDX-MS data. A.T. developed docking of Sgl1 substrates. A.T. and R.C.R. analyzed the docking data. J.K. and T.C. re-synthesized Hits 1, 9, and 15 under supervision of I.O., N.P.S., M.D.P., and M.V.A. contributed intellectual and strategic input. M.D.P. and M.V.A. supervised work and provided funding support. N.P.S. and M.V.A. drafted the initial manuscript with contributions from A.T., J.E.B., and I.O. N.P.S., M.D.P., and M.V.A. edited the final manuscript. All authors approved the final manuscript.

## Competing interests

M.D.P. is a Co-Founder and Chief Scientific Officer (CSO) of MicroRid Technologies Inc. whose goal is to develop new antifungal agents of therapeutic use. The remaining authors declare no competing interests.
