## [Peer Review File · Nature Communications]

Structure and inhibition of Cryptococcus neoformans sterylglucosidase to develop antifungal agentsREVIEWERS' COMMENTS

Reviewer #1 (Remarks to the Author):

I have previously reviewed the manuscript and find that the revision has addressed the majority of my concerns. The greatest concerns were the potential impact of the findings and proof that the site-directed variants were well folded. Importantly, the point is made that the inhibitors cause the accumulation of erg-glc, although less than that effected by genetic deletion of the Sgl1. New data has also been added to show efficacy of Cn Sgl1 inhibition in a mouse model of infection and that the inhibitor shows similar growth defects as the Sg1 deletion strain in *neoformans*. Addressing the concern about misfolding, the authors do show that there is diminished activity (rather than no activity) for the variants, although I note that to be most thorough, CD spectra of more than one of the mutants would be preferable in case it is a folded population giving activity. There are a couple of additional points which the authors may consider:

Page: 5- In the previous version of the manuscript, this reviewer had suggested that the authors include a structure-based sequence alignment to show that the large loop deletion (723-777) is not conserved in orthologs. They have included the alignment but have not addressed this point in the manuscript when the deletion is described (line 132). Additionally, it would be useful to delineate this segment in the alignment itself.

Figure 3- Unless I missed it, I don't see a description of the inset in new Figure 3e. Also in 3e, it is not easy to tell the black from grey and a different color choice would be better. Lastly, I think this data would be easier to look at in tabular form.

Reviewer #2 (Remarks to the Author):

In this work, the authors have conducted studies on the enzyme sterylglucosidase-1 (Sgl1) in the human pathogenic fungus, *Cryptococcus neoformans*. Sgl1 is known to hydrolyze ergosterol-3-beta-D-glucoside (Erg-glc), and its deletion causes *C. neoformans* to become non-pathogenic. Thus, Sgl1 is a promising molecular target and its characterization could lead to new treatments for fungal infections.

In the work presented here, the authors have identified key differences in the overall structure as well as the active site of Sgl1 when compared with bacterial and human glucosylceramidases. They have also identified important amino acid residues required for interactions with the substrate. The authors conducted a screen of 50,000 compounds and identified 3 hit compounds that inhibited Sgl1 activity and also caused the accumulation of Erg-glc in *C. neoformans* cells. In a mouse model of cryptococcosis, one of the inhibitors reduced cryptococcal burden in the brain compared to the vehicle control. Finally, docking studies revealed how the compounds interact with key residues in Sgl1's active site.

The data presented are of high quality and are well-presented. The data provide new insights into the structural architecture of Sgl1, and lay the groundwork for the generation of potent inhibitors of Sgl1 that can be developed into therapeutic or preventive treatments for treating life-threatening fungal infections.

Specific comments:

1. For this reviewer, the authors have addressed all the concerns that were raised in the previous submission. Grammatical errors have been corrected. Additional discussion points that were requested have been added. The addition of Supplemental Figure 8, showing that the hit compounds don't inhibit Cerezyme activity against its native substrate, provides strong evidence that the compounds specifically inhibit the fungal enzyme.

2. While the new data in the revised manuscript, demonstrating the growth effects of the inhibitors and the in vivo efficacy of Hit 1, are very interesting, they raise a couple of questions:

a. In their previous work (reference 11), the authors have shown that the *sgl1* deletion mutant was very similar to the wild type strain in its growth characteristics, even under stress conditions (e.g., pH stress, oxidative stress, and phagocytosis by macrophages). However, in the present study, limiting oxygen and some nutrients, causes a dramatic effect on the growth of the *sgl1* deletion mutant. A little more discussion on the biological significance of this effect and how it relates to the cellular functions of Sgl1 in *C. neoformans* would be useful.

b. Previously, the authors have reported that the deletion of Sgl1 renders *C. neoformans* non-pathogenic. The new data in this work show that under specific conditions, the deletion of Sgl1 also kills *C. neoformans* cells, and that pharmacological inhibition of Sgl1 under the same conditions also kills them. This raises the question – is the reduced brain burden observed in the in vivo study, that was seen with Hit 1, due to reduced virulence of *C. neoformans* cells or due to their killing. Some discussion on this issue would be useful to include.

3. Minor points:

a. The word “broad-spectrum” in the abstract may be a little premature. The studies described here are focused on *C. neoformans*, so to say that this work develops Sgl1 as a broad-spectrum target seems a little inaccurate.

b. In Fig. 4H, the colors used to outline the bars in the bar graphs are difficult to see, and it is difficult to differentiate between Hit 1 and Hit 15 results.

Pereira de Sa et al – Reviewer response

Reviewer #1 (Remarks to the Author):

I have previously reviewed the manuscript and find that the revision has addressed the majority of my concerns. The greatest concerns were the potential impact of the findings and proof that the site-directed variants were well folded. Importantly, the point is made that the inhibitors cause the accumulation of erg-glc, although less than that effected by genetic deletion of the Sgl1. New data has also been added to show efficacy of Cn Sgl1 inhibition in a mouse model of infection and that the inhibitor shows similar growth defects as the Sgl1 deletion strain in neoformans. Addressing the concern about misfolding, the authors do show that there is diminished activity (rather than no activity) for the variants, although I note that to be most thorough, CD spectra of more than one of the mutants would be preferable in case it is a folded population giving activity.

There are a couple of additional points which the authors may consider:

Page: 5- In the previous version of the manuscript, this reviewer had suggested that the authors include a structure-based sequence alignment to show that the large loop deletion (723-777) is not conserved in orthologs. They have included the alignment but have not addressed this point in the manuscript when the deletion is described (line 132). Additionally, it would be useful to delineate this segment in the alignment itself.

Response: We apologize that we did not address this point more clearly in the manuscript. A brief discussion that the deleted region appears to be specific of Cn Sgl1 was added to the results section (see page 5). We also highlighted the deleted region in the alignment presented in the supplementary material (see supplementary figure 3).

Figure 3- Unless I missed it, I don't see a description of the inset in new Figure 3e. Also in 3e, it is not easy to tell the black from grey and a different color choice would be better. Lastly, I think this data would be easier to look at in tabular form.

Response: The inset description was added to the legend of figure 3e. We decided to keep the data in a graphic format but changed the color to black (erg-glc) and red (res-glp) as the values are all available in the source data and the main point of reduced or non-reduced activity is captured in the figure format.

Reviewer #2 (Remarks to the Author):

In this work, the authors have conducted studies on the enzyme sterylglucosidase-1 (Sgl1) in the human pathogenic fungus, *Cryptococcus neoformans*. Sgl1 is known to hydrolyze ergosterol-3-beta-D-glucoside (Erg-glc), and its deletion causes *C. neoformans* to become non-pathogenic. Thus, Sgl1 is a promising molecular target and its characterization could lead to new treatments for fungal infections.

In the work presented here, the authors have identified key differences in the overall structure as well as the active site of Sgl1 when compared with bacterial and human glucosylceramidases. They have also identified important amino acid residues required for interactions with the substrate. The authors conducted a screen of 50,000 compounds and identified 3 hit compounds that inhibited Sgl1 activity and also caused the accumulation of Erg-glc in *C. neoformans* cells. In

a mouse model of cryptococcosis, one of the inhibitors reduced cryptococcal burden in the brain compared to the vehicle control. Finally, docking studies revealed how the compounds interact with key residues in Sgl1's active site.

The data presented are of high quality and are well-presented. The data provide new insights into the structural architecture of Sgl1 and lay the groundwork for the generation of potent inhibitors of Sgl1 that can developed into therapeutic or preventive treatments for treating life-threatening fungal infections.

Specific comments:

1. For this reviewer, the authors have addressed all the concerns that were raised in the previous submission. Grammatical errors have been corrected. Additional discussion points that were requested have been added. The addition of Supplemental Figure 8, showing that the hit compounds don't inhibit Cerezyme activity against its native substrate, provides strong evidence that the compounds specifically inhibit the fungal enzyme.

2. While the new data in the revised manuscript, demonstrating the growth effects of the inhibitors and the in vivo efficacy of Hit 1, are very interesting, they raise a couple of questions:

a. In their previous work (reference 11), the authors have shown that the sgl1 deletion mutant was very similar to the wild type strain in its growth characteristics, even under stress conditions (e.g., pH stress, oxidative stress, and phagocytosis by macrophages). However, in the present study, limiting oxygen and some nutrients, causes a dramatic effect on the growth of the sgl1 deletion mutant. A little more discussion on the biological significance of this effect and how it relates to the cellular functions of Sgl1 in *C. neoformans* would be useful.

Response: We now added more discussion about the biological significance of this effect (please see page 14).

b. Previously, the authors have reported that the deletion of Sgl1 renders *C. neoformans* non-pathogenic. The new data in this work show that under specific conditions, the deletion of Sgl1 also kills *C. neoformans* cells, and that pharmacological inhibition of Sgl1 under the same conditions also kills them. This raises the question – is the reduced brain burden observed in the in vivo study, that was seen with Hit 1, due to reduced virulence of *C. neoformans* cells or due to their killing. Some discussion on this issue would be useful to include.

Response: Yes, this is a possibility. We now discuss this issue in the discussion section (please see page 14).

3. Minor points:

a. The word "broad-spectrum" in the abstract may be a little premature. The studies described here are focused on *C. neoformans*, so to say that this work develops Sgl1 as a broad-spectrum target seems a little inaccurate.

Response: The term broad-spectrum was removed from abstract.

b. In Fig. 4H, the colors used to outline the bars in the bar graphs are difficult to see, and it is difficult to differentiate between Hit 1 and Hit 15 results.

Response: Agreed. The colors were revised for easier identification of results from each hit.